# Is Carmustine Wafer Implantation in Progressive High-Grade Gliomas a Relevant Therapeutic Option? Complication Rate, Predictors of Complications and Onco-Functional Outcomes in a Series of 53 Cases

**DOI:** 10.3390/cancers16203465

**Published:** 2024-10-12

**Authors:** Grigorios Gkasdaris, Julien Berthiller, Jacques Guyotat, Emmanuel Jouanneau, Clémentine Gallet, David Meyronet, Laure Thomas, Stéphanie Cartalat, Antoine Seyve, Jérôme Honnorat, François Ducray, Thiebaud Picart

**Affiliations:** 1Department of Neurosurgery, Hôpital Neurologique Pierre Wertheimer, Hospices Civils de Lyon, 69677 Bron, France; jacques.guyotat@gmail.com (J.G.); emmanuel.jouanneau@chu-lyon.fr (E.J.); clementine.gallet@chu-lyon.fr (C.G.); 2Department of Research and Clinical Epidemiology—Public Health, Hospices Civils de Lyon, 69677 Bron, France; julien.berthiller@chu-lyon.fr; 3Faculty of Medicine, University Claude Bernard Lyon I, 69100 Villeurbanne, France; david.meyronet@chu-lyon.fr (D.M.); antoine.seyve@chu-lyon.fr (A.S.); jerome.honnorat@chu-lyon.fr (J.H.); francois.ducray@chu-lyon.fr (F.D.); 4Cancer Initiation and Tumoral Cell Identity Department, Cancer Research Centre of Lyon (CRCL) INSERM 1052, CNRS 5286, 69008 Lyon, France; 5Department of Neuropathology, Groupement Hospitalier Est, Hospices Civils de Lyon, 69677 Bron, France; 6Department of Neuro-Oncology, Hôpital Neurologique Pierre Wertheimer, Hospices Civils de Lyon, 69677 Bron, France; laure.thomas@chu-lyon.fr (L.T.); stephanie.cartalat01@chu-lyon.fr (S.C.); 7MELIS Institute—Team Synaptopathies and Autoantibodies, INSERM U1314, UMR CNRS 5284, 69677 Bron, France

**Keywords:** 1,3-bis (2-chloroethyl)-1-nitrosourea, BCNU, carmustine wafer, postoperative complication, progressive glioblastoma, progressive high-grade glioma, surgical site infection, survival analysis

## Abstract

**Simple Summary:**

Intracavitary chemotherapy by Carmustine wafer implantation represents a therapeutic option for the management of high-grade gliomas both at diagnosis and at progression. However, this strategy is very controversial as it can lead to potential complications and previous studies have raised doubts regarding its efficacy in terms of oncological outcomes. Moreover, the results associated with Carmustine wafer implantation have been more frequently studied at diagnosis than at progression. Therefore, the aim of the present study was to precisely identify the predictors of complications and onco-functional outcomes in a series of 53 patients with a high-grade glioma surgically managed at progression with implantation of Carmustine wafers. These analyses will help to better identify and select the patients who are the best candidates to receive Carmustine wafers at progression and to guide intraoperative and postoperative management.

**Abstract:**

**Background/Objectives**: The aim was to determine the complication rate and the predictors of complications and survival in high-grade glioma surgically managed at progression with implantation of Carmustine wafers. **Methods**: A retrospective series of 53 consecutive patients operated on between 2017 and 2022 was built. **Results**: The median age was 55 ± 10.9 years. The rates of global and infectious complications were 35.8% and 18.9%, respectively. In multivariate analysis, patients with a preoperative neurological deficit were more prone to develop a postoperative complication (HR = 5.35 95% CI 1.49–19.26, *p* = 0.01). No predictor of infectious complication was identified. In the grade 4 glioma subgroup (n = 44), progression-free and overall survival (calculated starting from the reresection) reached 3.95 months, 95% CI 2.92–5.21 and 11.51 months, 95% CI 9.11–17.18, respectively. Preoperative KPS > 80% (HR = 0.97 95% CI 0.93–0.99, *p* = 0.04), Gross Total Resection (HR = 0.38 95% CI 0.18–0.80, *p* = 0.01), and 3-month postoperative KPS > 80% (HR = 0.35 95% CI 0.17–0.72, *p* = 0.004) were predictors of prolonged overall survival. **Conclusions**: Surgical resection is a relevant option in high-grade gliomas at progression, especially in patients with a preoperative KPS > 80%, without preoperative neurological deficit, and amenable to complete resection. In patients elected for surgery, Carmustine wafer implantation is associated with a high rate of complications. It is consequently critical to closely monitor the patients for whom this option is chosen.

## 1. Introduction

High-grade gliomas (HGGs) are the most common and severe primary malignant brain tumors in adults. Maximal safe resection, when feasible, represents the first step of management and its quality is a predictor of onco-functional prognosis, independent of the molecular features [1,2,3,4]. If the diagnosis of glioblastoma is confirmed, the postoperative treatment is well codified and now consists of concomitant and adjuvant radio-chemotherapy with temozolomide [4,5], associated with the use of tumor-treating fields [6]. Despite multimodal management, glioblastoma progression is unfortunately observed almost systematically after a median interval of around 15 months following diagnosis, according to the most recent clinical data [6,7], and is generally local [8]. The management of glioblastoma at progression is still not consensual [7,8,9,10]. Reresection seems to provide an increase in the overall survival (OS) of approximately 6 months and an improvement in the quality of life if complete [11]. A still-ongoing randomized trial was designed in order to better assess the value of reresection in terms of OS compared to the best oncological treatment (NCT06283927). Whether a reresection is performed or not, the second line of the oncological treatment most frequently consists of an association of nitrosourea and antiangiogenic drugs [7,12,13,14]. Unfortunately, still few patients survive beyond two years after the progression [6,7,15,16].

The implantation of Carmustine (1,3-bis(2-chloroethyl)-1-nitrosourea) biodegradable wafer (CWI) in the surgical bed is possible after HGG (re)resection. This strategy was initially developed with the goal of offering therapeutic coverage between the surgery and the onset of the adjuvant treatment. Additionally, CWI has the advantage of leading to the local delivery of an antineoplastic agent, whose diffusion is consequently not limited by the blood–brain barrier, unlike agents administered by other routes [17,18]. Although encouraged by some neurosurgical societies [17], this option is controverted in the setting of newly diagnosed HGGs as it seems to provide only a modest improvement in the OS [19,20] and is above all likely to preclude enrollment in clinical trials [7]. Conversely, CWI is more readily considered at progression, given its favorable impact on survival and the limited therapeutic options in these patients [10,18,19]. The rate of surgical site infection (SSI) following CWI in newly diagnosed HGGs reaches 4–28% [19,20,21,22,23,24,25,26,27,28,29,30] and other complications such as brain edema [20,21,26,27,31] or surgical bed cysts are frequently reported [22,32,33]. The occurrence of these side effects in HGGs operated at progression, although reported in a clinical trial in 1995 [18], were not assessed in recent series. Therefore, the aim of the present study was to analyze the rate and the predictors of surgical complications, and the onco-functional outcomes, in patients with an HGG diagnosed in accordance with the 2021 WHO Classification of Tumors of the Central Nervous System and surgically managed with CWI at progression. The ultimate goal was to ease the selection of the patients who are the best candidates to receive CWI in this context.

## 2. Materials and Methods

### 2.1. Patient Selection

The database from the Hospices Civils de Lyon pharmacy was screened to retrospectively identify consecutive patients surgically managed with CWI for an HGG between 1 January 2017 and 31 December 2022. CWI was never proposed for newly diagnosed but rather only at progression, for the reasons exposed above. In all cases, the indication of reresection with CWI was validated during the tumor board meeting, on the basis of the guidelines from the French Neurosurgical Society [17]. All surgical procedures were conducted under neuronavigation guidance with the goal of performing a maximal safe resection of the contrast-enhanced part of the tumor. The use of intraoperative tools such as 5-ALA fluorescence-guided resection or intraoperative motor monitoring was at the discretion of the operating surgeon. An intraoperative extemporaneous examination was systematically asked to rule out any differential diagnosis (radionecrosis notably) and CWI was performed only if HGG progression was confirmed by the pathologist.

All patients meeting the following inclusion criteria were included: (1) age ≥ 18 years; (2) available pathological results from both resections; (3) available postoperative MRI scan performed in the 72 h following the reresection (including T1-weighted with and without contrast, FLAIR, and diffusion sequences); and (4) clinico-radiological follow-up ≥6 months following the reresection.

### 2.2. Data Collection

Medical records, available imaging, and operative reports were thoroughly reviewed for each patient.

Demographic data, medical history including history of immunosuppression (such as splenectomy, leucopenia, immunosuppressor intake, diabetes, chronic alcoholism) and chronic infectious site (such as chronic urinary or dental infection), clinical status, radiological features, surgical characteristics (extent of reresection, number of implanted Carmustine wafers, duration of the hospitalization following reresection, mean corticosteroids dosage during the 3 postoperative weeks), surgical complications, integrated pathological diagnosis according to the 2021 WHO classification [34], molecular features, treatment modalities (prior and after progression), date of clinical and/or radiological progression following reresection according to the RANO criteria [35], if any, and the date of last follow-up or death were collected.

The extent of reresection was assessed on the MRI scan performed in the 72 h following the reresection. Gross Total Resection (GTR) was defined by a resection of 100% of the contrast-enhanced part of the glioma and Sub-Total Resection (STR) was defined by an incomplete resection with a resection rate >90% of the contrast-enhanced part.

Early postoperative complications included immediate postoperative neurological worsening, surgical site hematoma, and superficial and deep SSI. The rate of early rehospitalization in the two weeks following patient discharge was also assessed. The constitution of a postoperative peri-cavitary edema was noted. Delayed complications included cyst formation and general and neurological worsening, linked or not to the constitution of a peri-cavitary edema, assessed 3 months and 6 months postoperatively.

### 2.3. Survival

The survival analysis was conducted only in patients with a grade 4 glioma. The progression-free survival (PFS) was defined as the time between tumor reresection and the date of progression (as defined above) or the date of the last follow-up if the patient did not recur. The OS was defined as the time between tumor reresection and the date of death. For surviving patients, this interval was censored at the date of the last follow-up.

### 2.4. Statistical Analysis

Categorical variables were expressed as a number (n) and percentage. Quantitative variables were expressed as mean ± standard deviation when the distribution was normal. The hypothesis of normal distribution of quantitative variables was tested using the Shapiro test and graphically confirmed with a histogram.

Categorical variables were compared using the chi-2 test or Fisher’s exact test when the conditions of application of the chi-squared test were not met. Quantitative variables were compared between groups using the Student’s *t* test after verification of equality of variances when data were normally distributed and with the nonparametric test of Wilcoxon when the hypothesis of normality of distribution was not verified.

A logistic regression was conducted in order to identify risk factors for early and infectious postoperative complications, using a backward stepwise approach, first as a univariate analysis and second as a multivariate analysis including significant variables in the univariate analysis (*p*-value < 0.05 level), as well as variables defined as pertinent by the scientific board for the clinical interpretation of the results, such as sex and age.

Description of PFS and OS was estimated by the Kaplan–Meier product limit method and the effect of different parameters was assessed using the log rank test.

Prognostic factors were assessed using the semi-parametric Cox model after verification of the proportional hazard hypothesis, using a backward stepwise approach, first as a univariate analysis, then as a multivariate analysis including significant results from the univariate analysis (*p*-value < 0.05 level with mortality), as well as a variable defined as pertinent by the scientific board for the clinical interpretation of the results, such as age and gender. The best multivariate model both in the logistic regression and Cox model was determined using the Akaike information criterion.

The statistical tests were bilateral, and the level of significance was set to 5% (*p* < 0.05). Statistical analyses were conducted using SAS version 9.4 (SAS Institute Inc., Cary, NC, USA). Figures were built with GraphPad version 5 (La Jolla, CA, USA).

### 2.5. Standard Protocol Approvals and Registrations

Study design and manuscript organization were guided by the STROBE statement on cohort studies. This study was conducted in accordance with local and international ethical standards, as well as the 1964 Helsinki Declaration and its later amendments. All patients provided informed consent for tumor sample inclusion in the Hospices Civils de Lyon biological resource center and gave informed consent for the retrospective extraction of their clinical data. This study was approved by the Institutional Review Board of the Hospices Civils de Lyon (IRB 24-5106).

## 3. Results

The screening of the Hospices Civils de Lyon pharmacy database identified 54 patients surgically managed with CWI for an HGG between 1 January 2017 and 31 December 2022. After the exclusion of one duplicate, all the 53 remaining patients met the inclusion criteria and were considered.

### 3.1. Characteristics of the Patients

The characteristics of the whole series and of patients who had grade 4 gliomas are separately detailed in Table 1. For the whole series, the mean age at diagnosis was 55 ± 10.9 years (range, 24–80 years). The sex ratio of male/female was 1.78. Regarding the medical history, 10 (18.9%) patients were immunocompromised (leucopenia n = 5, immunosuppressor drug intake n = 2, chronic alcoholism n = 2, splenectomy n = 1) and 4 (7.5%) patients had a chronic infectious site (n = 2 urinary, n = 2 dental).

#### 3.1.1. Glioma Characteristics and Management at Diagnosis

Radiologically, gliomas were located in the frontal (n = 21, 39.6%), parietal (n = 11, 20.8%), temporal (n = 18, 34.0%), or occipital (n = 3, 5.7%) lobes (Figure 1a). In 28 (52.8%) patients, the tumor was located in the right hemisphere. Of note, 39 (90.7%) patients were right-handed.

Histologically, most of the tumors corresponded to glioblastoma *IDH* wild type (n = 43, 81.1%), 20 (41.6%) of which displayed an *EGFR* amplification. The remaining cases were all IDH mutant tumors, including one (1.9%) astrocytoma grade 4, two (3.8%) astrocytomas grade 3, and seven (13.2%) oligodendrogliomas grade 3 (Figure 1b). In the whole series, 36 (90%) tumors were *TERT*-mutant (including 32 glioblastomas *IDH* wild type and 4 oligodendrogliomas) and the *MGMT* promotor was methylated in 33 (75%) cases.

Regarding the management at diagnosis, 44 (83.0%) patients had an extended resection (GTR or STR) while 9 (17.0%) patients had a partial resection or a biopsy (Figure 1c). The adjuvant oncological treatment consisted of a standard Stupp radio-chemotherapy regimen with six adjuvant temozolomide cycles (n = 36, 67.9%). Eight (15.1%) patients received a standard Stupp radio-chemotherapy regimen with more than six adjuvant temozolomide cycles. Finally, nine (17.0%) patients received another treatment that consisted of radiotherapy only (n = 1), PCV regimen (procarbazine, CCNU, and vincristine) (n = 1), or radiotherapy and PCV regimen (n = 6) (Figure 1d).

#### 3.1.2. Clinical Status and Management at Progression

The diagnosis of glioma progression was based only on radiological parameters in 37 (69.8%) patients and on radiological parameters associated with a neurological worsening in the remaining cases. The median preoperative KPS score was 90.0 ± 10.0%. The preoperative neurological examination identified a neurological deficit in 20 (37.7%) patients (detailed description below in Table 1) and signs of elevated intracranial pressure in 2 (3.8%) patients. Six (11.3%) patients had epileptic seizures (Figure 1e).

Regarding prescriptions, seven (13.2%) patients were taking long-term antibiotic therapy (detailed below in Table 1). Additionally, 24 (45.3%) patients were taking corticosteroids preoperatively, at a mean dose of 124.5 ± 175.5 mg eq hydrocortisone.

The reresection was performed after a mean delay of 38.7 ± 49.6 months after the first resection and consisted of GTR (n = 26, 49.1%) or STR (n = 27, 50.9%) (Figure 1f). Surgery was guided by 5-ALA fluorescence in 32 (60.4%) patients. The lateral ventricle was opened in 23 (48.9%) of cases. On average, 7.4 ± 0.8 Carmustine wafers were implanted.

### 3.2. Postoperative Course and Surgical Complications

On average, patients stayed 8.6 ± 3.0 days in the hospital. Postoperatively, 33 (62.3%) patients received corticosteroids at a mean dose of 141.5 ± 131.5 mg eq hydrocortisone (Table 2).

#### 3.2.1. Early Postoperative Complications and Rehospitalization Rate

In the whole series, 20 (37.7%) patients developed at least one early postoperative complication. Two (3.8%) patients developed a post-surgical hematoma that was managed surgically (n = 1) or conservatively (n = 1). The immediate postoperative examination found a new or increased neurological deficit in nine (17.3%) patients (detailed below Table 2). Additionally, four (7.5%) patients experienced postoperative seizures. Ten (18.9%) patients developed an SSI (Figure 1g). Their characteristics are described in Table 3. The most frequent causal bacterium was Meticilline-sensible *Staphylococcus aureus* (n = 7, 70%). In all cases, there was a superficial infection, which was isolated (n = 1, 10.0%) associated with meningitis (n = 6, 60.0%), deep infection (n = 1, 10.0%), or both (n = 1, 10.0%).

After discharge, 19 (35.8%) patients were rehospitalized during the two first postoperative weeks for the management of an infectious complication (n = 10, 52.6%), a neurological worsening (n = 3, 15.8%), a hydrocephalus or a subdural hygroma (n = 2, 10.5%), a pseudo-meningocele (n = 2, 10.5%), epileptic seizures (n = 1, 5.3%), or headaches (n = 1, 5.3%).

#### 3.2.2. Predictors of Early and Infectious Postoperative Complications

In order to better identify patients at risk of early postoperative complications (as previously defined) or infectious complications only, a univariate ± multivariate analysis was carried out (Table 4).

A preoperative KPS > 80% predicted a significantly decreased risk of early postoperative complication (HR 0.19, 95% CI 0.06–0.65, *p* = 0.008) in univariate but not in multivariate analysis. However, according to the multivariate analysis, the presence of a preoperative neurological deficit was an independent predictor of an increased risk of early postoperative complication (HR 5.35, 95% CI 1.49–19.26, *p* = 0.01).

The univariate analysis did not identify any predictor of infectious complications.

#### 3.2.3. Delayed Complications and Functional Prognosis

In the 3 months following the reresection, 11 patients (20.8%) developed a peri-cavitary edema. In seven patients (13.2%), this edema caused neurological worsening, leading to the introduction of corticosteroids or an increase in the dose, if patients were already taking this prescription. Additionally, the 3-month postoperative MRI demonstrated the presence of a cyst in the surgical site in 25 (52.1%) patients (Figure 1g). This radiological feature was associated with neurological worsening in 8/25 (32%) patients.

From a functional point of view, 3 months postoperatively, 21 (43.8%) patients had a KPS ≤ 80% (Figure 1h). Twenty-one (43.8%) and nineteen (39.6%) respectively presented with a worsened general or neurological status, compared to the preoperative examination. Six months postoperatively, 25 (53.2%) patients had a worsened neurological status, compared to the preoperative examination. In univariate analysis, the predictors of neurological worsening were corticosteroids preoperative intake (uHR = 3.84 95% CI 1.35–10.97, *p* = 0.01) and 3-month radiological progression (uHR = 3.69 95% CI 1.21–11.24, *p* = 0.02) (Table 5).

### 3.3. Oncological Prognosis

#### 3.3.1. Postoperative Management and Survival Analysis in Patients Managed for a Grade 4 Glioma

Given the restricted number of patients managed for a grade 3 glioma (n = 9), only data related to patients with a grade 4 glioma are presented (n = 44). In this subgroup, after reresection, 39 (88.6%) patients received an adjuvant treatment that consisted most frequently of chemotherapy with the use of different drug combinations (bevacizumab n = 36, temozolomide n = 14, carboplatine n = 18, belustine n = 30). Three patients (6.8%) were included in a clinical trial and received targeted therapies or anti-PD1 drugs. Two (4.5%) patients had a third surgery later in the course of the disease. Finally, one (2.3%) patient was reradiated, and one (2.3%) patient was treated with TTFields.

At the end of the study, 42 (95.5%) patients with a grade 4 glioma were dead. In this subgroup, the PFS and the OS (calculated from the reresection) were respectively 3.95 months, 95% CI 2.92–5.21 and 11.51 months, 95% CI 9.11–17.18 (Figure 2a,b). Of note, the OS from the diagnosis was 37.47 months, 95% CI 14.37–118.07.

#### 3.3.2. Predictors of PFS and OS in Patients with a Grade 4 Glioma

In univariate analysis, predictors of prolonged PFS were GTR (unadjusted uHR 0.31 95% CI 0.15–0.62, *p* = 0.001) and 3-month postoperative KPS > 80% (uHR 0.31 95% CI 0.14–0.69, *p* = 0.003), as detailed in Table 6 and Figure 2c. Conversely, 3-month neurological worsening (uHR 2.33, 95% CI 1.19–4.59, *p* = 0.01) was a predictor of decreased PFS.

In univariate analysis, predictors of prolonged OS were preoperative KPS > 80% (uHR 0.97, 95% CI 0.93–1.00, *p* = 0.04), GTR (uHR 0.40, 95% CI 0.20–0.83, *p* = 0.01), and 3-month postoperative KPS > 80% (uHR 0.37, 95% CI 0.19–0.73, *p* = 0.004) Conversely, 3-month neurological worsening (uHR 1.97, 95% CI 1.01–3.87, *p* = 0.04) was a predictor of decreased OS. According to the multivariate analysis, preoperative KPS > 80% (aHR 0.97, 95% CI 0.93–0.99, *p* = 0.04), GTR (aHR 0.38, 95% CI 0.18–0.80, *p* = 0.01), and 3-month postoperative KPS > 80% (aHR 0.35, 95% CI 0.17–0.72, *p* = 0.004) were independent predictors of prolonged OS (Figure 2d,e).

## 4. Discussion

In the present series of 53 patients with an HGG surgically managed with CWI at progression, the global rate of early surgical complications, SSI, and early rehospitalization were 37.7%, 18.9%, and 35.8%, respectively. The global rate of complications was in line with that observed in previously published series and significantly higher compared to that associated with CWI at diagnosis [36]. Additionally, in a series of 63 glioblastomas at progression, the rate of wound-healing complications and SSI reached 14.2% [26].

Conversely, hemorrhagic complications were rare and concerned less than 5% of patients, as previously observed [37], but they seemed to be more frequent than in patients who underwent resection without CWI [37]. Finally, postoperative epileptic seizures were also rare (7.5%). In a series of 55 cases, the rate of postoperative seizures was very close (9%). The occurrence of this event was not influenced by the number of implanted Carmustine wafers and was essentially observed in patients who developed other surgical complications [38].

### 4.1. Identification of Patients at Risk of Early Surgical Complications Following Reresection of HGG Associated with CWI

In the present study, patients with a preoperative KPS < 80% or a neurological deficit were predisposed to develop an early surgical complication after HGG reresection associated with CWI. Insofar as immediate postoperative neurological worsening is considered an early surgical complication, this association is not surprising. Indeed, patients with a preoperative neurological deficit obviously have tumors located close to or within eloquent areas and are logically more prone to have an increased neurological deficit postoperatively.

Regarding surgical considerations, ventricle opening was not associated with a higher rate of complications, consistently with observations made in HGG managed at diagnosis and at progression with CWI [27,39,40]. However, it seems critical to carefully repair a ventricular defect, if any, before CWI [40]. In a series mixing glioblastomas at diagnosis and at progression, a high number of Carmustine wafer (n = 8) was associated with an increased risk of adverse events [36]. In the present series, most of the patients (67.3%) received ≥8 Carmustine wafers, which probably limited the power of the analysis to uncover a similar association. Yet, in a series of HGG with CWI at diagnosis, the number of implants was not a predictor of complications [41].

None of the variables (including notably preoperative immunosuppression, the pre- and postoperative doses of corticosteroids, the number of implanted Carmustine wafers, and the length of hospital stay) included in the dedicated univariate analysis were identified as predictors of SSI. Predictors of SSI after CWI at progression were studied in a non-recent series of 32 glioblastomas. The time since the previous resection and the radiation dose were higher in patients who developed SSI than in patients who did not, only in univariate analysis. Additionally, all patients who received vancomycin for surgical prophylaxis (n = 3) developed an SSI [25]. Of note, in the present series, only one of the 10 patients who developed an SSI received vancomycin. All the others received third-generation cephalosporins.

Taken together, these data indicate that the risk of early surgical complications in this population is easier to predict than that of SSI. However, an immediate postoperative neurological worsening, which represents an early surgical complication, can evidently recover in the months following surgery. Thus, the 3-month postoperative functional status, which represents a critical parameter to consider for patient management as it conditions the quality of life, was further analyzed.

### 4.2. Functional Outcomes after the Reresection of HGG Associated with CWI

In the present series, the rate of immediate neurological worsening reached 17%. Three months postoperatively, about 40% of patients had a worsened neurological status compared to the preoperative status. These rates are particularly high but consistent with those observed in a series of HGGs surgically managed at progression [42,43]. Preoperative corticosteroid intake was a predictor of 3-month postoperative neurological worsening, but not the presence of a preoperative deficit. This discrepancy could result from a lack of statistical power. Nevertheless, it is also possible that the rate of preoperative neurological deficit was underestimated, notably because patients with tumors located in eloquent areas (i.e., at high risk of postoperative deficit) were wrongly considered deficit-free because they took corticosteroids preoperatively, which masked a neurological deficit. While an immediate postoperative deficit directly results from the surgical management, a 3-month postoperative deficit can also result from other factors such as the occurrence of a complication, the constitution of a peri-cavitary edema or a surgical bed cyst, or tumor progression.

The rate of peri-cavitary edema (13.2%) was high compared to previous series [19,25]. About 50% of the patients developed a surgical bed cyst in the present series. Cysts were associated with a neurological worsening in about 30% of patients. In a series of 43 patients who received CWI for an HGG at diagnosis or at progression, the rate of one-month postoperative bed cysts was close (58% of cases) [33]. Cysts were more frequently present when CWI was performed at progression than at diagnosis [32,33]. Other risk factors for cyst formation are age ≥60 years, incomplete resection, and the implantation of more than eight Carmustine wafers [22,33]. As observed in the present series, most cysts are asymptomatic [22,32], but they can also sometimes cause signs of elevated intracranial pressure, which are promptly responsive to corticosteroids [33] and rarely justify more invasive management [32].

### 4.3. Oncological Outcomes Following Reresection of HGG Associated with CWI

For the 44 grade 4 glioma patients included in the present series, the PFS and the OS were respectively 4 months and 11.5 months. It is important to note that this group is composed of “elite HGG patients”, eligible for several resections, and thus have a better prognosis than HGG patients who are not [21,44,45,46]. In previous series of HGGs treated with CWI at progression, OS ranged from 7 months to 13 months [18,19,28,30,31,47,48,49,50,51].

Yet, these data are difficultly comparable as pathological diagnoses were based on different versions of the WHO classification of tumors of the central nervous system, and the proportion of true glioblastomas may vary from one series to another.

The extent of reresection and 3-month postoperative general and neurological status were strong predictors of survival outcomes. In a series mixing glioblastomas surgically managed with CWI at progression ± at diagnosis, the extent of resection [28,36] and the preoperative KPS [28] were also identified as strong predictors of OS. According to a French retrospective multicentric study including 559 HGG at progression, predictors of survival were rather related to the previous oncological management. Indeed, temozolomide and radiation administered before and after CWI, bevacizumab administered before CWI, and a longer delay between the first and the second resection were associated with a longer OS [47].

In another series including 56 HGG at progression, a transient increase in peri-cavitary hypersignal in T2-FLAIR weighted sequences, which probably emanated from an inflammatory process, was identified as a favorable prognosis factor [31].

Importantly, the data obtained from the series of HGG at diagnosis and at progression both indicate that CWI is more effective when the *MGMT* promoter is methylated [50,52,53,54]. In the present series, the *MGMT* promoter status was not a predictor of survival but a methylated status was frequently observed (72.5% of patients with grade 4 glioma), thus limiting the strength of the analysis regarding this parameter.

### 4.4. Impact of CWI on the Onco-Functional Balance of Patients with Progressive HGG

Because of the absence of a controlled group of HGG surgically managed without CWI at progression, it was not possible to determine to what extent CWI modulated the onco-functional prognosis. A controlled randomized trial showed that CWI significantly increased the OS compared to the placebo (31 weeks vs. 23 weeks), and was not associated with a higher rate of complications. Yet, this trial was led in 1995 and was consequently not in accordance with the current WHO classification of tumors and the standards of care, including notably the Stupp protocol [18]. According to a meta-analysis, 1-year OS rates in patients surgically managed for a grade 4 glioma at progression without and with CWI were 40% and 42%, respectively. For 2-year OS rates, the corresponding values were 13% and 17% [19]. These results indicate that CWI confers only a very modest advantage in this indication [19,36,49].

In HGGs, infectious complications are generally more frequent at progression than at diagnosis [55,56]. Additionally, a previous series showed that CWI in glioblastoma at progression is associated with a higher risk of postoperative complications (especially wound healing and infection) than reresection only [36]. In the present series, the occurrence of early surgical complications was not a predictor of survival. Consistently, the higher postoperative infection rate in glioblastoma patients at diagnosis ± at progression with CWI compared to patients without did not affect survival [21,26]. In accordance with these findings, it was established that SSI occurring after the first resection of a glioblastoma represents a very severe event, leading to treatment discontinuation and a significant decrease in the OS [57], although these results remain controversial [24,58]. Conversely, SSI after the second resection was not associated with a decrease in OS [57]. Yet, SSI results in rehospitalization, a new surgery with a possible ablation of the bone flap, and prolonged antibiotic therapy. Thus, its impact on the quality of life must not be underestimated.

### 4.5. Limitations of the Study

The main limitation of the present study is inherent in its retrospective design. The power of the statistical analyses was limited because of the relatively restricted size of the series, and this may affect the generalizability of the results to a broader population of patients with high-grade gliomas. Finally, no control group of HGG patients managed surgically without CWI could be created from our database. However, a few series specifically studied the surgical results associated with CWI in HGG at progression. Moreover, patients were managed homogeneously, and pathological diagnoses were revised in accordance with the 2021 WHO classification, thus limiting confounding factors and providing useful indications to optimize the surgical management of HGG at progression. Few patients (n = 9) of the present series had a grade 3 glioma, and 5 of them (55.6%) were alive at the end of the study, thus limiting the relevance of conducting a survival analysis in this subgroup. Future studies would be required to more specifically assess the relevance of CWI in grade 3 gliomas. Yet, all patients were followed for at least 18 months after the reresection, which seems to be sufficient to comprehensively capture the occurrence of surgical complications.

## 5. Conclusions

In patients with HGG, surgery at progression must be considered as promptly as possible, with the goal of achieving the largest possible resection while avoiding neurological worsening. The question of whether to associate CWI or not is not so easy to answer. Indeed, CWI is associated with a high rate of complication while its benefits in terms of survival are very limited.

Patient selection for CWI at reresection should be performed very carefully. Patients who are more likely to tolerate CWI at reresection typically have a non-altered preoperative general status (KPS > 80%), no preoperative neurological deficit, a tumor displaying a highly methylated *MGMT* promotor, and are amenable to GTR. Regarding the management, it seems important to inform the selected patients preoperatively of the high rate of complications associated with this strategy. The surgical closure (including that of the ventricle if opened) and the wound monitoring must be performed very carefully. Finally, given the risk of postoperative seizures, it also appears important to make sure that an antiepileptic prophylaxis is optimally administered.

## Figures and Tables

**Figure 1 cancers-16-03465-f001:**
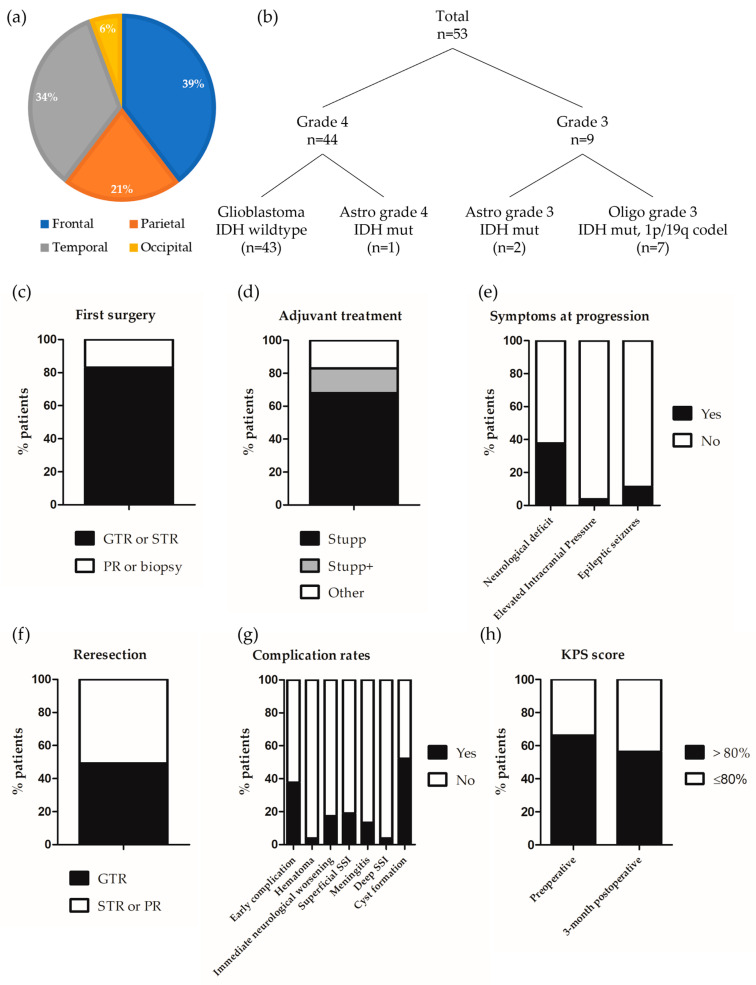
Characteristics of the series. Graphical representation of (**a**) Tumor location; (**b**) Integrated diagnosis; (**c**) Extent of the first surgery; (**d**) Adjuvant treatment; (**e**) Symptoms at progression; (**f**) Extent of the reresection; (**g**) Complication rates; and (**h**) 3-month postoperative KPS score, preoperative KPS score is indicated for comparison.

**Figure 2 cancers-16-03465-f002:**
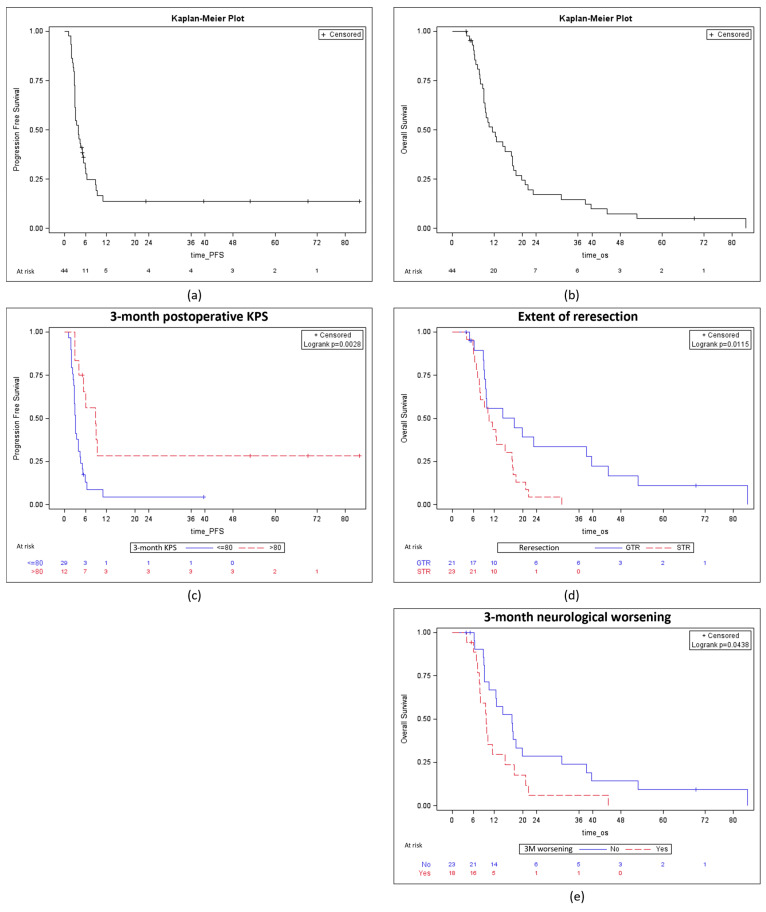
Survival analysis (**a**) Kaplan–Meier analysis for PFS; (**b**) Kaplan–Meier analysis for OS; (**c**) Kaplan–Meier analysis for PFS according to the 3-month postoperative KPS; (**d**) Kaplan–Meier analysis for OS according to the extent of reresection; (**e**) Kaplan–Meier analysis for OS according to 3-month postoperative neurological status.

**Table 1 cancers-16-03465-t001:** Clinical and surgical features of the whole series (n = 53) and of patients managed for a grade 4 glioma (n = 44).

	Whole Series (n = 53)	Grade 4 Gliomas Only (n = 44)
**DEMOGRAPHIC DATA**		
**Sex**		
Female	19 (35.8)	19 (43.2)
Male	34 (64.2)	25 (56.8)
**Age** (years) mean ± SD	55 ± 10.9	56 ± 11.4
**Age categories**		
≤60 years	35 (66.0)	26 (59.1)
>60 years	18 (34.0)	18 (40.9)
**MEDICAL HISTORY**		
**Immunosuppression**		
Yes	10 (18.9)	9 (20.5)
No	43 (81.1)	35 (79.5)
**Chronic infectious site**		
Yes	4 (7.5)	4 (9.1)
No	49 (92.5)	40 (90.9)
**LOCATION AND HISTO-MOLECULAR CHARACTERISTICS OF THE GLIOMA**		
**Location**		
Frontal	21 (39.6)	14 (31.8)
Parietal	11 (20.8)	10 (22.7)
Temporal	18 (34.0)	17 (38.7)
Occipital	3 (5.7)	3 (6.8)
**Integrated diagnosis**		
Glioblastoma *IDH* wild type	43 (81.1)	43 (97.7)
Astrocytoma grade 4 *IDH* mutant	1 (1.9)	1 (2.3)
Astrocytoma grade 3 *IDH* mutant	2 (3.8)	0 (0)
Oligodendroglioma grade 3 *IDH* mutant, 1p/19q co-deleted	7 (13.2)	0 (0)
***EGFR* status**		
Amplified	20 (41.6)	20 (46.5)
Non amplified	28 (58.3)	23 (53.5)
Missing	5	0
***TERT* status**		
Mutation (C228T or C250T)	36 (90.0)	32 (88.9)
Wild-type	4 (10.0)	4 (11.1)
Missing	13	8
***MGMT* status**		
Methylated	33 (75.0)	29 (72.5)
Unmethylated	11 (25.0)	11 (27.5)
Missing	9	4
**INITIAL MANAGEMENT**		
**Extension of the first surgery**		
GTR or STR	44 (83.0)	40 (90.9)
Partial resection or biopsy	9 (17.0)	4 (9.1)
**Adjuvant treatment**		
Stupp	36 (67.9)	35 (79.5)
Stupp +	8 (15.1)	8 (18.2)
Other	9 (17.0)	1 (2.3)
**CLINICAL DATA AT PROGRESSION**		
**Neurological deficit**		
Yes ^1^	20 (37.7)	18 (40.9)
No	33 (62.3)	26 (59.1)
**Elevated Intracranial Pressure**		
Yes	2 (3.8)	2 (4.5)
No	51 (96.2)	42 (95.5)
**Epileptic seizures**		
Yes	6 (11.3)	5 (11.4)
No	47 (88.7)	39 (88.6)
**Preoperative KPS score**		
>80%	35 (66.0)	29 (65.9)
≤80%	18 (34.0)	15 (34.1)
**Pre-operative antibiotics intake**		
Yes ^2^	7 (13.2)	7 (15.9)
No	46 (86.8)	37 (84.1)
**Pre-operative corticosteroids intake**		
Yes	24 (45.3)	22 (50.0)
No	29 (54.7)	22 (50.0)
**SURGICAL MANAGEMENT AT PROGRESSION**		
**Time between the first resection and the reresection** (months) mean ± SD	38.7 ± 49.6	24.7 ± 20.9
**Extent of the reresection**		
GTR	26 (49.1)	21 (47.7)
STR or partial resection	27 (50.9)	23 (52.3)
**Ventricular opening**		
Yes	23 (48.9)	22 (55.0)
No	24 (51.1)	18 (45.0)
Missing	6	4
**Number of implanted Carmustine wafers**		
<8	17 (32.7)	32 (72.7)
8	34 (65.4)	12 (27.3)
>8	1 (1.9)	0 (0)
Missing	1	0

Data are expressed as count (percentage) unless otherwise specified. ^1^ Language disorders (n = 9), lateral homonymous hemianopia (n = 8), motor deficit (n = 6), cognitive disorders (n = 4), ataxia (n = 1). Some patients had several of these symptoms. ^2^ Cotrimoxazole (n = 6), Amoxicillin + Metronidazole (n = 1).

**Table 2 cancers-16-03465-t002:** Postoperative course and complications the whole series (n = 53) and in patients managed for a grade 4 glioma (n = 44).

	Whole Series (n = 53)	Grade 4 Gliomas Only (n = 44)
**Length of hospital stay** (day) mean ± SD	8.6 ± 3.0	8.6 ± 3.2
**Dose of corticosteroids during the 3 postoperative weeks** (eq mg hydrocortisone) mean ± SD	141.5 ± 131.5	151.5 ± 133.5
**Post-surgical hematoma**		
Yes	2 (3.8)	1 (2.3)
No	51 (96.2)	43 (97.7)
**Immediate neurological worsening**		
Yes ^1^	9 (17.3)	9 (20.9)
No	43 (82.7)	34 (79.1)
Missing	1	1
**Superficial SSI**		
Yes	10 (18.9)	7 (15.9)
No	43 (81.1)	37 (84.1)
**Meningitis**		
Yes	7 (13.2)	4 (9.1)
No	46 (86.8)	40 (90.9)
**Deep SSI**		
Yes	2 (3.8)	1 (2.3)
No	51 (96.2)	43 (97.7)
**Early rehospitalization**		
Yes	19 (35.8)	14 (31.8)
No	34 (64.2)	30 (68.2)
**Cyst formation**		
Yes	25 (52.1)	19 (48.7)
No	23 (47.9)	20 (51.3)
Missing	5	5
**3-month postoperative KPS**		
>80%	27 (56.2)	23 (56.1)
≤80%	21 (43.8)	18 (43.9)
Missing	5	3
**3-month postoperative general or neurological worsening**		
Yes ^2^	23 (47.9)	21 (56.1)
No	25 (52.1)	18 (43.9)
Missing	5	5

Data are expressed as count (percentage) unless otherwise specified. ^1^ Motor deficit (n = 5), language disorders (n = 3), sensitive deficit (n = 1). ^2^ Decreased KPS (n = 21) ± neurological worsening (n = 19), included motor deficit (n = 12), language disorders (n = 5), lateral homonymous hemianopia (n = 3), sensitive deficit (n = 2). Some patients had a combination of several symptoms.

**Table 3 cancers-16-03465-t003:** Characteristics of the patients who developed a surgical site infection (n = 10).

	Wound Infection	Meningitis	Deep Surgical Site Infection	Microbiological Agent ^1^	Sex	Age (Years)	Immunosuppression	Chronic Infectious Site	Tumor Location ^2^	Integrated Diagnosis ^3^	Extension of the First Resection ^4^	Adjuvant Treatment ^5^	Preoperative Corticoids	Preoperative Antibiotics	Preoperative KPS (%)	Preoperative Neurological Deficit	Extent of the Reresection ^4^	Lateral Ventricle Opening	Implanted Carmustine Wafers	Length of Hospital Stay (Days)	Postoperative Dose of Corticosteroids ^6^
1	+	-	-	MSSA	M	38	-	-	P	GBM	STR	S	-	-	80	+	GTR	+	8	10	0
2	+	+	-	STREP	M	52	-	-	F	O3	STR	PCV	-	-	90	-	GTR	-	6	14	0
3	+	+	-	MSSA	M	74	-	-	T	GBM	STR	S	-	-	90	-	GTR	-	6	8	220
4	+	+	-	NA	M	46	+	-	F	GBM	STR	S	+	+	90	-	GTR	+	7	6	240
5	+	+	-	MSSA	M	52	-	-	T	O3	PR	PCV-RT	-	-	80	+	STR	-	8	7	0
6	+	+	-	MSSA	F	65	-	-	F	GBM	STR	S	+	-	80	+	STR	+	8	15	200
7	+	+	-	EBC	M	57	-	-	P	GBM	STR	S	+	-	90	+	STR	NA	6	22	0
8	+	-	-	MSSA	M	47	-	-	O	GBM	STR	S+	+	-	90	+	STR	-	7	6	248
9	+	-	+	MSSA + ECC	F	53	-	-	P	GBM	STR	S	+	-	90	-	STR	-	8	7	0
10	+	+	+	MSSA + CBA	M	48	-	-	P	A3	STR	S	+	-	80	-	STR	NA	8	7	200

^1^ CBA = *Cutibacterium acnes*, EBC = *Enterobacter cloacae*, ECC = *Enterococcus Casseliflavus*, MSSA = Meticilline-Sensible *Staphylococcus aureus*, STREP = *Streptococcus* spp. ^2^ F = Frontal, O = Occipital, P = Parietal, T = Temporal. ^3^ A3 = Astrocytoma grade 3, GBM = glioblastoma, O3 = Oligodendroglioma grade 3. ^4^ PR = Partial Resection, STR = Sub-Total Resection. ^5^ PCV = Procarbazine + CCNU + Vincristine, RT = Radiotherapy, S = Stupp, S+ = Stupp + additional temozolomide cycles. ^6^ Expressed in eq mg hydrocortisone. NA = Not Available. + means present and - means absent.

**Table 4 cancers-16-03465-t004:** Predictors of early postoperative and infectious complications.

Factor	Early Postoperative Complications				Infectious Complications	
	Univariate Analysis		Multivariate Analysis		Univariate Analysis	
	HR [95% CI]	*p*	HR [95% CI]	*p*	HR [95% CI]	*p*
Sex (male)	0.93 [0.29–3.01]	0.91	1.09 [0.30–3.94]	0.89	1.38 [0.31–6.12]	0.67
Age > 60 years	0.38 [0.10–1.39]	0.14	1.00 [0.95–1.07]	0.82	0.42 [0.08–2.23]	0.31
Immunosuppression	0.72 [0.16–3.20]	0.67			0.42 [0.05–3.76]	0.44
Tumor location (parietal)	2.40 [0.54–10.69]	0.25			3.43 [0.61–19.40]	0.16
Integrated diagnosis of HGG	0.75 [0.03–17.51]	0.64			0.19 [0.01–3.39]	0.25
Preoperative corticosteroids	1.00 [0.97–1.02]	0.75			0.98 [0.94–1.02]	0.31
Preoperative antibiotics	0.68 [0.12–3.19]	0.67			0.69 [0.07–6.43]	0.74
Preoperative KPS (> 80%)	0.19 [0.06–0.65]	**0.008**	0.74 [0.05–10.00]	0.81	0.97 [0.91–1.04]	0.41
Preoperative neurological deficit	1.62 [0.48–5.40]	0.43	5.35 [1.49–19.26]	**0.01**	0.52 [0.10–2.76]	0.44
Extent of reresection (STR)	2.17 [0.69–6.87]	0.18			1.57 [0.39–6.37]	0.53
Lateral ventricle opening	0.73 [0.22–2.45]	0.61			0.57 [0.12–2.72]	0.48
Number of implanted Carmustine wafers	0.98 [0.67–1.43]	0.91			0.88 [0.54–1.42]	0.60
Length of hospital stay	1.12 [0.92–1.36]	0.25			1.21 [0.97–1.51]	0.08
Postoperative dose of corticosteroids	1.01 [0.99–1.04]	0.22			0.99 [0.96–1.02]	0.50

Significant *p*-values are indicated in bold characters.

**Table 5 cancers-16-03465-t005:** Predictors of 3-month postoperative neurological worsening (univariate analysis).

Factor	Neurological Worsening	
	HR [95% CI]	*p*
Sex (male)	2.50 [0.96–6.49]	0.06
Age > 60 years	2.50 [0.91–6.82]	0.07
Tumor location (parietal)	0.65 [0.18–2.37]	0.90
Integrated diagnosis of HGG	4.76 [0.59–35.80]	0.15
Preoperative corticosteroids	3.84 [1.35–10.97]	**0.01**
Preoperative KPS (>80%)	0.46 [0.13–1.59]	0.22
Preoperative neurological deficit	0.95 [0.36–2.51]	0.92
Extent of reresection (STR)	0.85 [0.31–2.32]	0.75
Lateral ventricle opening	0.56 [0.21–1.44]	0.22
Number of implanted Carmustine wafers	1.05 [0.82–1.36]	0.69
Early postoperative complication	1.17 [0.43–3.16]	0.76
3-month radiological progression	3.69 [1.21–11.24]	**0.02**

Significant *p*-values are indicated in bold characters.

**Table 6 cancers-16-03465-t006:** Predictors of PFS and OS in patients with a grade 4 glioma.

Factor	PFS		OS			
	Univariate Analysis		Univariate Analysis		Multivariate Analysis ^1^	
	HR [95% CI]	*p*	HR [95% CI]	*p*	HR [95% CI]	*p*
Sex (male)	054 [0.27–1.05]	0.07	0.92 [0.48–1.77]	0.81	1.24 [0.62–2.46]	0.54
Age	0.93 [0.46–1.86]	0.83	1.02 [0.99–1.05]	0.28	1.02 [0.99–1.05]	0.28
MGMT methylation	0.60 [0.27–1.29]	0.19	1.15 [0.54–2.41]	0.71		
Extent of first resection: STR or GTR	0.98 [0.34–2.77]	0.96	1.04 [0.37–2.96]	0.94		
Preoperative KPS > 80%	1.02 [0.51–2.03]	0.95	0.97 [0.93–1.00]	**0.04**	0.97 [0.93–0.99]	**0.04**
Preoperative neurological deficit	1.02 [0.53–1.99]	0.95	1.97 [1.01–3.87]	0.05		
Extent of reresection: GTR	0.31 [0.15–0.62]	**0.001**	0.40 [0.20–0.83]	**0.01**	0.38 [0.18–0.80]	**0.01**
Number of implanted Carmustine wafer	0.92 [0.71–1.20]	0.53	0.95 [0.75–1.19]	0.64		
5-ALA use	1.06 [0.54–2.06]	0.86	1.21 [0.64–2.29]	0.56		
Lateral ventricle opening	0.43 [0.21–0.87]	0.02	0.67 [0.35–1.29]	0.23		
Early postoperative complication	1.02 [0.53–1.97]	0.94	1.45 [0.76–2.77]	0.27		
3-month neurological worsening	2.33 [1.19–4.59]	**0.01**	1.97 [1.01–3.87]	**0.04**	1.79 [0.86–3.76]	0.07
3-month postoperative KPS > 80%	0.31 [0.14–0.69]	**0.003**	0.37 [0.19–0.73]	**0.004**	0.35 [0.17–0.72]	**0.004**

^1^ Adjusted on age and sex. Significant *p*-values are indicated in bold characters.

## Data Availability

The data presented in this study are available upon reasonable request from the corresponding authors (Grigorios Gkasdaris, Thiébaud Picart).

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
