# Peer review of "Is Carmustine Wafer Implantation in Progressive High-Grade Gliomas a Relevant Therapeutic Option? Complication Rate, Predictors of Complications and Onco-Functional Outcomes in a Series of 53 Cases"

_cancers, 2024, doi:10.3390/cancers16203465_

Round 1
Reviewer 1 Report
Comments and Suggestions for Authors
The study by Gkasdaris et al. is a well-conducted analysis of predictors of complications related to Carmustine wafer implantation in high-grade gliomas. The authors meticulously analyzed data from 53 patients, categorized according to the 2021 WHO classification, and presented their findings clearly.
To further enhance the work, I would suggest adding a more detailed discussion of the recent literature. Specifically, while the authors cite the works of Champeaux-Depond (2023), these studies are not addressed in the discussion section. Incorporating and analyzing these findings in the context of the current results could strengthen the discussion and provide a broader perspective on the subject.
Author Response
The study by Gkasdaris et al. is a well-conducted analysis of predictors of complications related to Carmustine wafer implantation in high-grade gliomas. The authors meticulously analyzed data from 53 patients, categorized according to the 2021 WHO classification, and presented their findings clearly.
We would like to warmly thank Reviewer 1 for his/her positive feedback.
Comment 1: To further enhance the work, I would suggest adding a more detailed discussion of the recent literature. Specifically, while the authors cite the works of Champeaux-Depond (2023), these studies are not addressed in the discussion section. Incorporating and analyzing these findings in the context of the current results could strengthen the discussion and provide a broader perspective on the subject.
Response 1: We thank reviewer 1 for this relevant remark. Indeed, the works of Champeaux-Depond are very relevant and robust as they are based on wide and multicentric series of patients. One the study published in 2023 was carried out in patients with high-grade gliomas who received carmustine wafer at recurrence (ref [47]). The results of this study are already discussed as follows: “According to a French retrospective multicentric study including 559 HGG at progression, predictors of survival were rather related to the previous oncological management. Indeed, temozolomide and radiation administered before and after CWI, bevacizumab administered before CWI and a longer delay between the first and the second resection were associated with a longer OS [47].” To the best of our knowledge, all the key findings from this study are discussed in our manuscript.
The second study from Champeaux-Depond et al, also robust, was carried out in patients who received Carmustine at diagnosis and not at recurrence, unlike to our study. Consequently, the studied population is not the same and it is not easy to integrate these results in the discussion which is only focused on patients managed at recurrence. Nevertheless, this second article (ref [45]) is also already cited as follows: “It is important to note that this group is composed of “elite HGG patients”, eligible to several resections, which have thus a better prognosis than HGG patients who are not [20,44–46]”.
Reviewer 2 Report
Comments and Suggestions for Authors
The manuscript is generally well-structured, but certain sections could improve:
It would be beneficial to include more details on the statistical analyses performed, particularly regarding the logistic regression and Cox model analyses.
While the results are presented clearly, the inclusion of additional visual aids such as adding more figures would enhance the reader's understanding.
With only 53 patients included in the analysis, the sample size is relatively small. This limits the statistical power of the study and may affect the generalizability of the results to a broader population of patients with high-grade gliomas.
The long-term follow-up time may not capture all problems and survival data. Longer follow-up is needed to assess therapy efficacy and safety.
Limitations must be disclosed for scientific honesty integrity therefore add one subheading.
Overall, the manuscript presents valuable findings that contribute to the understanding of Carmustine wafer implantation in high-grade gliomas. Addressing these comments will enhance the quality and impact of the study.
Author Response
The manuscript is generally well-structured, but certain sections could improve:
We thank reviewer 2 for his/her positive feedback.
Comment 1: It would be beneficial to include more details on the statistical analyses performed, particularly regarding the logistic regression and Cox model analyses.
Response 1: We thank reviewer 2 for raising this point. The section “statistical analysis” was completed as follows:
“A logistic regression was conducted in order to identify risk factors for early and infectious postoperative complications, using a backward stepwise approach, first as a univariate analysis and second as a multivariate analysis including significant variables in the univariate analysis (p-value <0.05 level) as well as variables defined as pertinent by the scientific board for the clinical interpretation of the results, as sex and age.”
“Prognostic factors were assessed using the semi-parametric Cox model after verification of the proportional hazard hypothesis, using a backward stepwise approach, first as a univariate analysis, then as a multivariate analysis including significant results from the univariate analysis (p-value <0.05 level with mortality) as well as variable defined as pertinent by the scientific board for the clinical interpretation of the results such as age and gender. The best multivariate model both in logistic regression and cox model, was determine using the Akaike information criterion.”
Comment 2: While the results are presented clearly, the inclusion of additional visual aids such as adding more figures would enhance the reader's understanding.
Response 2: We thank reviewer 2 for this suggestion. The addition of visual aids is obviously always helpful for the reader. So, a figure with 8 panels was created to summarize the key characteristics of the series.
Comment 3: With only 53 patients included in the analysis, the sample size is relatively small. This limits the statistical power of the study and may affect the generalizability of the results to a broader population of patients with high-grade gliomas.
Response 3: We totally agree with this remark which was partly discussed in the limitation section of the manuscript (paragraph 4.4). To be clearer regarding this limitation, the sentence was modified as follows: “The power of the statistical analyses was limited because of the relatively restricted size of the series and may affect the generalizability of the results to a broader population of patients with high-grade gliomas.”
Comment 4: The long-term follow-up time may not capture all problems and survival data. Longer follow-up is needed to assess therapy efficacy and safety.
Response 4: We thank reviewer 2 for this remark. It is true that a restricted follow-up may not capture all problems and survival data. Yet, at the end of the present study, 65.5% (42/44) of the patients with a grade 4 glioma were dead. Consequently, we do not think that the follow-up was too short to prevent to analyze survival in this subgroup. However, 5/9 patients with a grade 3 glioma were still alive, explain why we did not conduct any survival analysis in this subgroup. Nevertheless, all patients were followed at least 18 months postoperatively, which is sufficient to comprehensively capture surgical complications (whose description was the main goal of the study). The limitation section was rephrased as follows: “Few patients (n=9) of the present series had a grade 3 glioma and 5 of them (55.6%) were alive at the end of the study, thus limiting the relevance to conduct a survival analysis in this subgroup. Future studies would be required to more specifically assess the relevance of CWI in grade 3 gliomas. Yet, all patients were followed during at least 18 months after the reresection, which seems to be sufficient to comprehensively capture the occurrence of surgical complications”
Comment 5: Limitations must be disclosed for scientific honesty integrity therefore add one subheading.
Response 5: We agree that it is of paramount importance to discuss the limitations of the study and we believe that scientific honesty must always be respected. Yet, in the initial manuscript, there was already a section entitled “Limitations of the study” (paragraph 4.4). This paragraph was completed in accordance with the previous comments from Reviewer 2.
Overall, the manuscript presents valuable findings that contribute to the understanding of Carmustine wafer implantation in high-grade gliomas. Addressing these comments will enhance the quality and impact of the study.
We thank reviewer 2 for this positive feedback.
Reviewer 3 Report
Comments and Suggestions for Authors
I would like to start by congratulating the authors for their work. An analysis that can help better identify and select the patients who are the best candidates to receive carmustine wafers upon progression, and guide intraoperative and postoperative management, is a step forward in addressing high-grade gliomas that give the patient an unfavorable prognosis.
- The introduction should be enriched with a more detailed argumentation on the implantation of biodegradable carmustine wafers.
- Additionally, the rationale should be better explained, given that it is mentioned as providing only modest improvement.
- I would recommend adding explanatory histograms in paragraph 3.1.1.
- It would be better to create a limitations section to reformulate the conclusions of the work.
- There is a formatting error in the first reference.
Author Response
I would like to start by congratulating the authors for their work. An analysis that can help better identify and select the patients who are the best candidates to receive carmustine wafers upon progression, and guide intraoperative and postoperative management, is a step forward in addressing high-grade gliomas that give the patient an unfavorable prognosis.
We thank Reviewer 3 for his/her positive feedback.
Comment 1: The introduction should be enriched with a more detailed argumentation on the implantation of biodegradable carmustine wafers.
Response 1: We thank reviewer 3 for this suggestion. The rational of Carmustine use was strengthen, as follows: “This strategy was initially developed with the goal to offer a therapeutic coverage between the surgery and the onset of the adjuvant treatment. Additionally, CWI has the advantage to lead to the local delivery of an antineoplastic agent, whose diffusion is consequently not limited by the blood-brain barrier, unlike agents administered by other routes [17,18]. »
Comment 2: Additionally, the rationale should be better explained, given that it is mentioned as providing only modest improvement.
Response 2: We thank Reviewer 3 for raising this point. It is true that, as indicated in the introduction, the improvement in the survival provided by Carmustine is modest in the context of newly-discovered glioblastomas but this population was not targeted in the present article. Therefore, it was added in the initial manuscript that CWI is more readily considered at progression, given the limited therapeutic options. Additionally, to strengthen the introduction, we added the following sentence: “Conversely, CWI is more readily considered at progression, given its favorable impact on survival and the limited therapeutic options in these patients.”
Comment 3: I would recommend adding explanatory histograms in paragraph 3.1.1.
Response 3: We thank reviewer 3 for this suggestion that match this of reviewer 2. We acknowledge that the addition of visual aids is obviously always helpful for the reader. So, a figure with 8 panels was created to summarize the key characteristics of the series.
Comment 4: It would be better to create a limitations section to reformulate the conclusions of the work.
Response 4: We thank reviewer 3 for this remark but are surprised as a limitation section already existed in the initial manuscript (4.4). In this paragraph, all the limitations we were able to identify were discussed: retrospective design, relatively small size of the series, impossibility to build a control cohort of patient surgically managed for a glioblastoma at progression without carmustine wafer implantation, and limited number of patients with a grade 3 glioma, limiting the relevance to conduct a survival analysis in this subgroup. Yet, following some comments of reviewer 2, this paragraph was completed as follows: “The power of the statistical analyses was limited because of the relatively restricted size of the series and may affect the generalizability of the results to a broader population of patients with high-grade gliomas.”, “Few patients (n=9) of the present series had a grade 3 glioma and 5 of them (55.6%) were alive at the end of the study, thus limiting the relevance to conduct a survival analysis in this subgroup. Future studies would be required to more specifically assess the relevance of CWI in grade 3 gliomas. Yet, all patients were followed during at least 18 months after the reresection, which seems to be sufficient to comprehensively capture the occurrence of surgical complications”
Comment 5: There is a formatting error in the first reference.
Response 5: We thank reviewer 3 for this remark. The formatting error is now fixed.